# Structure of human MUTYH and functional profiling of cancer-associated variants reveal an allosteric network between its [4Fe-4S] cluster cofactor and active site required for DNA repair

Carlos H. Trasviña-Arenas[1,7], Upeksha C. Dissanayake[2], Nikole Tamayo[1,3], Mohammad Hashemian[1,3], W. Jonathan Lin[1,3], Merve Demir[1,3], Nallely Hoyos-Gonzalez[1], Andrew J. Fisher [1,3,4], G. Andrés Cisneros [2,5] ✉, Martin P. Horvath [6] ✉ & Sheila S. David [1,3] ✉

MUTYH is a clinically important DNA glycosylase that thwarts mutations by initiating base-excision repair at 8-oxoguanine (OG):A lesions. The roles for its [4Fe-4S] cofactor in DNA repair remain enigmatic. Functional profiling of cancer-associated variants near the [4Fe-4S] cofactor reveals that most variations abrogate both retention of the cofactor and enzyme activity. Surprisingly, R241Q and N238S retained the metal cluster and bound substrate DNA tightly, but were completely inactive. We determine the crystal structure of human MUTYH bound to a transition state mimic and this shows that Arg241 and Asn238 build an H-bond network connecting the [4Fe-4S] cluster to the catalytic Asp236 that mediates base excision. The structure of the bacterial MutY variant R149Q, along with molecular dynamics simulations of the human enzyme, support a model in which the cofactor functions to position and activate the catalytic Asp. These results suggest that allosteric cross-talk between the DNA binding [4Fe-4S] cofactor and the base excision site of MUTYH regulate its DNA repair function.

Oxidative DNA damage and its repair are intimately linked to disease[1]. Arguably, the most studied oxidative DNA lesion is 8-oxo-7,8-dihydro-guanine (OG) which has a high miscoding potential due to its mimicry of thymine during DNA replication, leading to GC → TA transversion mutations. Improperly placed adenines within pro-mutagenic OG:A base pairs are removed by the adenine glycosylase MUTYH, as the first step in Base Excision Repair (BER). Subsequent action of downstream BER enzymes and the OG glycosylase, OGG1, complete the repair process to restore the G:C base pair. The impact of defective repair of OG:A lesions is underscored by the link between inherited *MUTYH* variants and colorectal cancer, a cancer susceptibility syndrome referred to as MUTYH-associated polyposis (MAP)[1–9]. MAP is defined as

[1]Department of Chemistry, University of California, Davis, CA, USA. [2]Department of Chemistry and Biochemistry, University of Texas at Dallas, Richardson, TX, USA. [3]Chemistry and Chemical Biology Graduate Program, University of California, Davis, CA, USA. [4]Department of Molecular and Cellular Biology, University of California, Davis, CA, USA. [5]Department of Physics, University of Texas at Dallas, Richardson, TX, USA. [6]School of Biological Sciences, University of Utah, Salt Lake City, UT, USA. [7]Present address: Research Center on Aging, Center for Research and Advanced Studies (CINVESTAV), Mexico City, Mexico. ✉e-mail: andres@utdallas.edu; martin.horvath@utah.edu; ssdavid@ucdavis.edu

an autosomal recessive disorder where inherited biallelic *MUTYH* mutations lead to multiple colorectal polyps with an increased likelihood of developing colorectal cancer[10]. Moreover, mutations in *MUTYH* are increasingly associated with other types of cancer including extraintestinal cancers such as breast, ovary, bladder, thyroid and skin cancers[11].

Many of the >2800 germline and 600 somatic mutations reported in *MUTYH* map in proximity to its metal cofactors: a [4Fe-4S] cluster and a Zn linchpin (Fig. 1)[2,10]. The [4Fe-4S] cluster is coordinated by four cysteine ligands within the N-terminal catalytic domain, while the Zn(II) ion is coordinated by three cysteine ligands in the interdomain connector region (IDC) and one histidine within the catalytic domain (Fig. 1)[12,13]. In previous work, we showed that the two metal cofactors are required for mediating DNA lesion engagement necessary for MUTYH base excision activity; however, the cofactors do not directly participate in base excision catalysis via redox chemistry or as Lewis acids[3,8,12,14].

The role of the [4Fe-4S] cluster cofactor in MUTYH and related glycosylases has been a topic of great interest. A loop made by two Cys residues that coordinate the [4Fe-4S] cluster, referred to as the iron-sulfur cluster loop, mediates electrostatic interactions with the DNA backbone that are critical for recognition of the lesion containing substrate[15,16]. In addition, the [4Fe-4S] cluster in MUTYH and related [4Fe-4S] cluster-containing BER glycosylases has been proposed to facilitate DNA lesion location through redox communication with other [4Fe-4S] cluster-containing enzymes[17]. This process takes advantage of DNA dependent redox cycling of the cofactor ([4Fe-4S]$^{2+/3+}$) to modulate the affinity of the repair enzyme for DNA, and thereby, influence the efficiency of pinpointing the lesion's location[17,18].

To provide insight into the impact of cancer associated variants (CAVs) and roles of the [4Fe-4S] cofactor in MUTYH function, we obtained the first crystal structure of human MUTYH in complex with DNA and defined structure-function relationships for 12 [4Fe-4S] associated CAVs. The majority of the CAVs within the [4Fe-4S] cluster motif cause complete loss of both cofactors, and lead to failure for both adenine glycosylase activity and lesion DNA recognition. Notably, R241Q and N238S CAVs disrupt a hydrogen bond network that bridges the [4Fe-4S] cluster to the active site comprised of residues Cys290 ([4Fe-4S] cluster ligand), Arg241, Asn238, and catalytic residue Asp236. We show using functional and computational approaches that this H-bond network mediates a structural communication between the [4Fe-4S] cluster and the active site that influences the proper positioning and protonation status of the catalytic Asp236, thereby altering catalytic efficiency. Our findings provide evidence of an allosteric

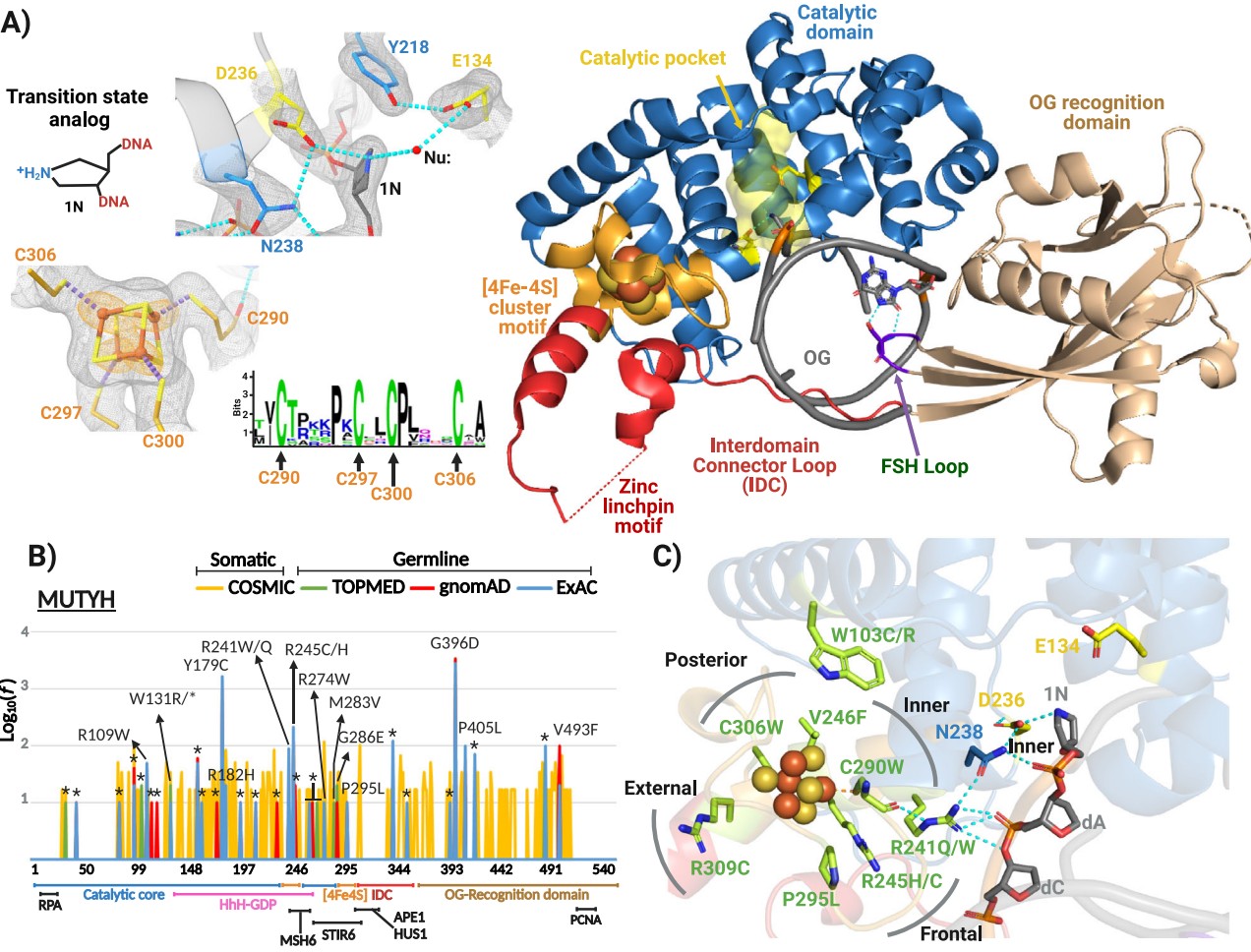

**Fig. 1 | Human MUTYH-TSAC structure and cancer-associated variants within the [4Fe-4S] cluster motif. A** Global structure (shown in ribbons) highlighting functional domains, catalytic residues and the [4Fe-4S] cluster. The simulated annealing composite omit (gray) and anomalous difference (orange) maps were contoured at 1.1- and 6-sigma, respectively. Functional domains and motifs are indicated in different colors. A logo sequence shows the conservation of the cysteine ligands across archaea, bacteria and eukaryote MutY/MUTYH. **B** Frequency and distribution of pathogenic germline and somatic mutations

mapped in MUTYH amino acid sequence. Domains and motifs are indicated with labels below amino acid numbering. Pathogenic somatic mutations (yellow) were obtained from the COSMIC database[23]. Pathogenic germline mutations were acquired from ExAC[86] (blue), gnomAD[87] (red) and TOPMED[88] (green) databases. Reported frequencies were log-normalized (log base 10). Nonsense mutations are indicated with asterisk. Protein-protein interaction sites are shown with black labels. **C** Structural mapping of cancer-associated variants within the [4Fe-4S] cluster motif.

**Table 1 | Metal analysis by Inductively Coupled Plasma-Mass Spectrometry (ICP-MS)**

| MUTYH | [4Fe-4S] load | Zn load |
|---|---|---|
| WT (MBP free) | 5.1 ± 0.1 | 1.2 ± 0.1 |
| WT | 4.4 ± 0.1 | 0.2 ± 0.1 |
| W103C | 0.0 | 0.0 |
| W103R | 0.0 | 0.0 |
| N238S | 4.1 ± 0.2 | 0.3 ± 0.1 |
| R241Q | 4.4 ± 0.1 | 0.1 ± 0.02 |
| R241W | 0.0 | 0.0 |
| R245H | 0.0 | 0.0 |
| R245C | 0.0 | 0.0 |
| V246F | 3.9 ± 0.5 | 0.2 ± 0.1 |
| C290W | 0.0 | 0.0 |
| P295L | 0.0 | 0.0 |
| C306W | 0.0 | 0.0 |
| R309C | 4.9 ± 0.3 | 0.2 ± 0.01 |

Values equal to 0 mean that the protein sample had Zn and Fe content equal to the blank. Source data are provided as a Source Data file.

network between the [4Fe-4S] cluster and the active site that supports DNA repair activity of MUTYH. The [4Fe-4S] cluster is not simply a bystander; CAVs break the allosteric network as evidenced by loss of function for R241Q and N238S, which retain structure, hold onto the metal cofactor, and bind DNA but nevertheless fail as adenine glycosylases.

## Results

### Crystal structure of human MUTYH

Mapping CAVs onto the structure of MUTYH provides the ability to predict and rationalize structural and functional impacts. The available structures of mammalian MUTYH are a truncated human MUTYH structure which lacks the C-terminal domain[19], two mouse MUTYH (mMutyh) structures with DNA and a mMutyh C-terminal fragment with PCNA[13]. The human and mouse homologs share 74% identity (Supplementary Fig. 1), and therefore appropriate analysis of CAVs necessitates a human MUTYH structure. We designed and validated an appropriate human MUTYH construct, and along with optimization of overexpression and purification, obtained high yields of pure recombinant human MUTYH protein (see "Methods" and Supplementary Fig. 2). The crystallization conditions were optimized from those reported for mMutyh[13] with a key difference being that we paired OG with the azaribose (1N) transition state analog (Fig. 1A), whereas the mMutyh structure contains tetrahydrofuran (THF, a product analog) across from OG. Crystals diffracted to high resolution (1.9 Å) with a synchrotron source and refinement yielded final R/Rfree values of 0.180/0.209.

In the MUTYH-TSAC (Fig. 1), MUTYH embraces the entire DNA helix with contacts mediated by the two functional domains in a manner similar to other MutY homologs[6,13,20,21]. Likewise, important residues within the catalytic pocket such as Asp236, Glu134, Tyr218, 1N and the OG recognition sphere (OG and Ser447) have similar positionings and interactions as found for murine and bacterial homologs[6,8,13] (Fig. 1A and Supplementary Fig. 3). The IDC region that connects the C-terminal OG recognition and N-terminal catalytic domains in mammalian MUTYH is distinct from other homologs by its longer length and harboring a $Zn^{2+}$ ion. We previously coined this region as the "Zn linchpin motif" to reflect its role in coordinating the activity of both domains to support MUTYH activity[3,12]. ICP-MS analysis of the recombinant human MUTYH used for crystallography showed that prior to crystallization the protein was fully loaded with Fe and Zn (Table 1). However, within the IDC region we were unable to observe

electron density for residues 324-347 in chain A and 319-346 in chain D (there are two copies of the MUTYH-TSAC in the asymmetric unit) nor did we observe electron density for the Zn and its coordination sphere. X-ray absorption spectra of the MUTYH-TSAC crystals lacked the ~9650-eV peak characteristic for Zn (Supplementary Fig. 4). These observations suggest Zn was lost during the crystallization process. CAVs are distributed throughout MUTYH and impinge on the many functional domains and motifs (Fig. 1B). Importantly, the residues corresponding to CAVs nearby the [4Fe-4S] cofactor are clearly defined by electron density. These CAV positions are highly but not absolutely conserved between mMutyh and MUTYH; for example, the residue corresponding to Arg309 in the human enzyme is a Tyr in mMutyh. Moreover, each CAV position is within 4.5 Å of a species-specific variation, further underscoring the need for a structure of the human enzyme.

### Structural mapping of cancer-associated variants on the MUTYH-TSAC structure reveal a structural interaction between the [4Fe-4S] cluster and the active site

CAVs surround the [4Fe-4S] cofactor in the human MUTYH-TSAC (Fig. 1C) underscoring the importance of this region in enzyme function. At the posterior face of the cofactor (as defined in Fig. 1C) are the CAVs C306W (cluster ligand), and V246F. At the frontal face are P295L, R245H/C and at the external face R309C. Finally, at the inner face which forms an intersection between the [4Fe-4S] cluster and the catalytic pocket are W103C/R along with another cluster cysteinyl ligand mutant, C290W, and R241Q/W.

Intriguingly, inspection of the MUTYH-TSAC crystal structure revealed an intricate hydrogen bonding network that spans the 20 Å distance between the [4Fe-4S] cluster and the adenine excision pocket (Fig. 2A). The connection involves four residues with strong evolutionary signals as shown by residue coupling correlation analysis (Fig. 2B), starting with the [4Fe-4S] cluster ligand Cys290 that H-bonds with Arg241, which in turn interacts with Asn238, and lastly this Asn residue H-bonds with the key catalytic residue Asp236 (Fig. 2C). In addition, the amide hydrogen of the side chain of the Asn238 also interacts with the 5' phosphate of the 1N nucleotide. These connections make use of the multivalent Arg241 and Asn238 residues: Arg241 adopts a C-like shape that enables the positioning of internal $NH_1^\varepsilon$ moiety of the guanidine group to donate a H-bond to the main chain carbonyl amide of Cys290. Likewise, the $NH_2^{\eta 2}$ group of Arg241 maintains the H-bond network, interacting with the carbonyl of Asn238 side chain and the phosphodiester backbone nucleotide two bases upstream of the 1N moiety. Remarkably, all of the residues in the implicated structural bridge between the [4Fe-4S] cluster and the active site are annotated as CAVs (including N238S and D236N mutations). Moreover, a similar H-bond network is evolutionarily conserved in EndoIII and MIG (Fig. 2E and Supplementary Fig. 5), suggesting that its functional significance is shared in [4Fe-4S] cluster containing HhH BER glycosylases. In addition to the CAVs studied herein, several others have been reported near the [4Fe-4S] cofactor in LOVD database[22] (R245S and R309H) as well as a predicted pathogenic somatic mutation in the COSMIC database[23] (R241L), further implicating this region as a "hotspot" for functional disruption.

### Functional assays revealed distinct sets of MUTYH variants localized near the [4Fe-4S] cofactor

We purified 11 of the MUTYH CAVs near the [4Fe-4S] cofactor along with N238S, to discern the functional impact of the amino acid variations (Fig. 3). Due to instability of the variant enzymes to MBP removal, all analyses were carried out with the MBP-MUTYH fusion protein (Supplementary Fig. 6) as previously reported[24,25]. Of note, ICP-MS analysis of MBP-free and MBP-fusion MUTYH showed above 4 nmol of Fe per nmol of MUTYH suggesting that forms of the recombinant proteins were fully loaded with the [4Fe-4S] cluster. In the case of Zn

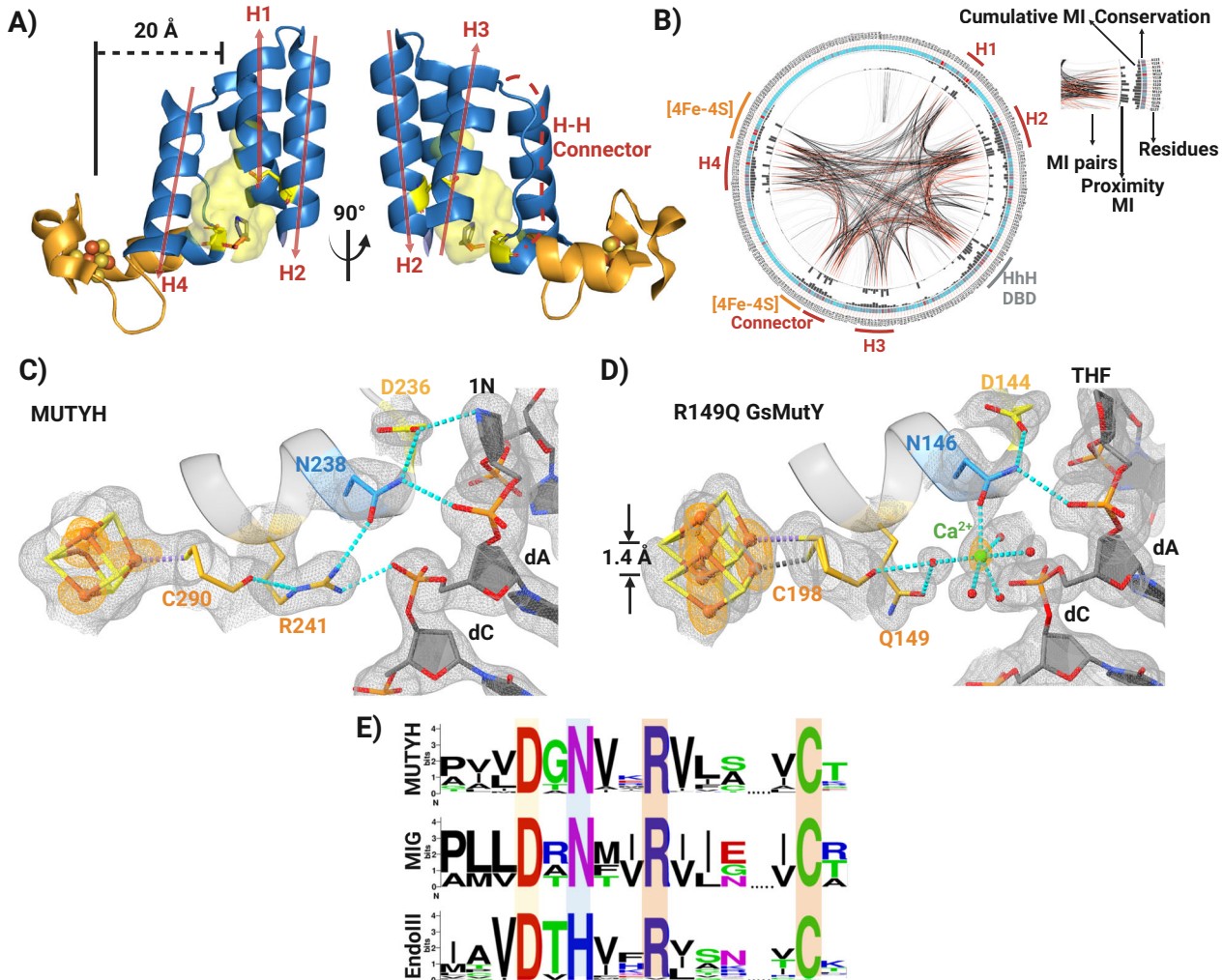

**Fig. 2 | Structural interplay between the [4Fe-4S] cluster and active site.**
**A** Structural composition of the catalytic pocket and proximity to the [4Fe-4S] cluster. The catalytic pocket comprises 4 α-helixes (H1-H4) and a Helix-Helix (H-H) connector. The H-H connector links the α-helix H3 with an α-helix, which is part of both the catalytic pocket, where Asn238 resides, and the [4Fe-4S] cluster. Catalytic Glu134 and Asp236 reside within the α-helix H2 and the H-H connector, respectively. **B** Mutual evolutionary relationships show a strong coevolutionary signal among the different structural components of the catalytic pocket and the [4Fe-4S] cluster motif of MUTYH/MutY proteins. Mutual Information (MI) detects positions within a multiple sequence alignment that are co-evolving to reveal evolutionary constraints imposed by structure or function. The protein sequence is presented as a circular plot, residue by residue. The different components taken into account by the coevolutionary analysis such as conservation, cumulative mutual information (MI), proximity MI, and MI pairs are depicted in the figure. **C** Residues implied in the structural connectivity between the [4Fe-4S] cluster and the active site. The H-bond network establishing this connectivity is shown in cyan dotted lines. The simulated annealing composite omit (gray) and anomalous difference (orange) maps were contoured at 1.1- and 6-sigma, respectively. **D** Disruption of the structural connectivity between the [4Fe-4S] cluster and the active site in the R149Q GsMutY-THF:OG crystal structure. This mutation shows alternative conformations for the [4Fe-4S] cluster with a displacement of 1.4 Å along with two different conformers of Cys 198. R149Q is the analogous R241Q mutation in MUTYH. The simulated annealing composite omit map (gray) was calculated to the 1.51 Å resolution limit and contoured at 1.0-sigma. **E** Conservation of the residues involved in the [4Fe-4S] cluster-active site structural connectivity among Helix-hairpin-Helix (HhH) DNA glycosylases that also contain the iron-sulfur cofactor (MUTYH/MutY, MIG and EndoIII).

loading, the MBP free MUTYH contained a full complement of the cofactor (1.2 nmol of Zn per/nmol of MUTYH; Table 1), while only 20% of the MBP-fusion MUTYH population retained Zn. The MBP is located at the N-terminal end of the MUTYH protein near the [4Fe-4S] cluster and Zn binding sites. Thus, the close proximity of the MBP to the metal binding sites might cause a propensity to lose the Zn coordination during the purification; a situation that is circumvented by the immediate MBP removal after Nickel affinity chromatography for the MBP-free MUTYH purification. Indeed, the Zn site in MUTYH is more labile than the [4Fe-4S] cluster, as it was also lost during the crystallization process.

A qualitative activity and binding screen showed that most of the MUTYH variants exhibited no adenine excision activity on a 30-bp OG:A-containing duplex (Duplex II, methods) nor any ability to bind the product analog-containing (OG:THF) DNA duplex (Fig. 3A, B, and Supplementary Figs. 7 and 8). Only V246F and R309C exhibited activity and duplex affinity at levels near that of WT MUTYH. Of particular note, R241Q and N238S retained the metal cofactor, bound the OG:THF-DNA duplex yet were inactive enzymes suggesting these two CAVs impact a previously unrecognized functionally required element (Fig. 3C).

The ability of the MUTYH CAVs to suppress DNA mutations in *E. coli* provided a means to assess activity in a cellular context. Specifically, rifampicin resistance assays measure the ability of MUTYH and variants, expressed as an MBP-MUTYH fusion protein, to suppress mutations in a *mutY* and *mutM* deficient GT100 *E. coli* strain (Fig. 3D, Supplementary Table 1)[12,25]. When cells were transformed with the *MUTYH*-containing

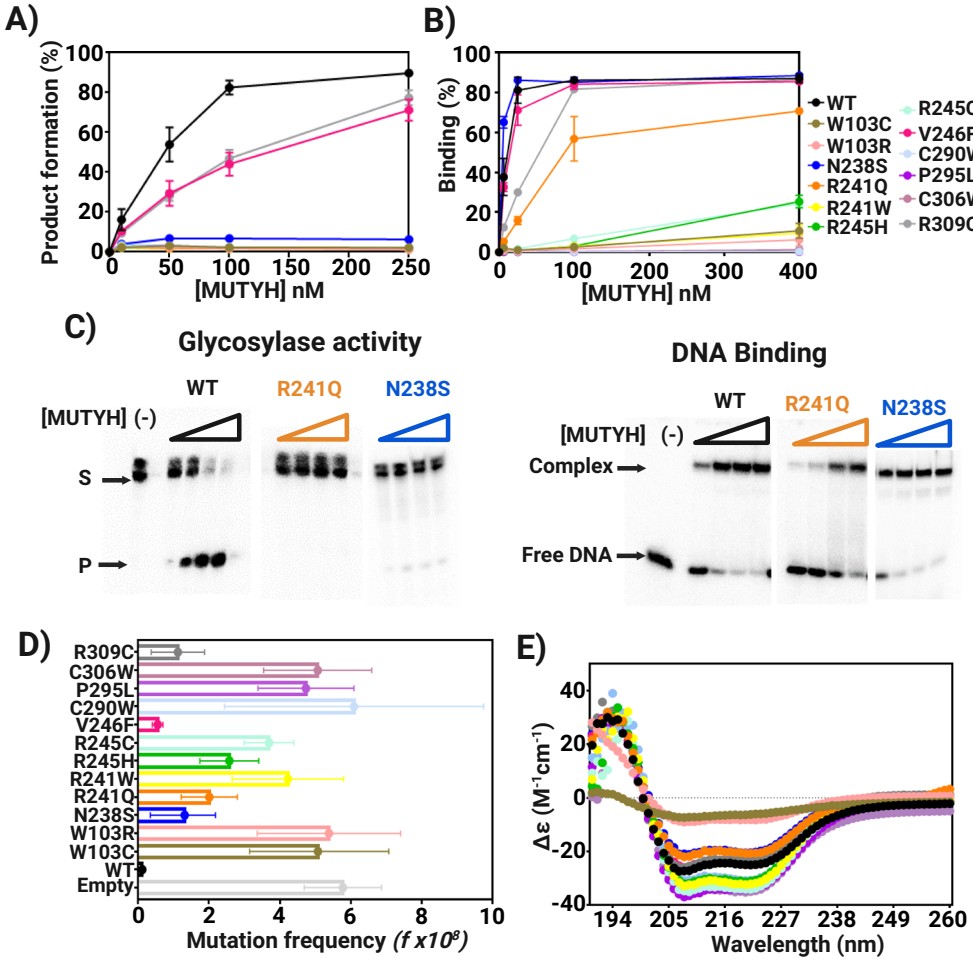

**Fig. 3 | Functional impact of [4Fe-4S] cluster cancer-associated variants of MUTYH. A** Total product formation as a function of [MUTYH] from adenine glycosylase assays of cancer-associated [4Fe-4S] cluster variants with OG:A-containing DNA substrates. The OG:A 30-bp duplex (20 nM) was incubated with increasing MBP-MUTYH concentrations (0–250 nM) for 1 h at 37 °C; see "Methods" for assay details **B** Percent DNA-bound as a function of [MUTYH] determined from electrophoretic mobility shift assays (EMSA) with cancer-associated variants. The THF:OG-containing 30-bp DNA duplex was titrated with increasing concentrations of MBP-MUTYH (0-400 nM absolute concentrations) and incubated for 20 min at 25 °C. Glycosylase and binding experiments were performed in triplicate with error

bars representing standard deviation from the average. **C** Representative storage phosphor autoradiograms of qualitative glycosylase and electrophoretic mobility shift assays (EMSA) with WT, R241Q and N238S MBP-MUTYH mutants. These experiments were carried out as described in Methods. **D** Mutation frequency measured using rifampicin resistance assay with GT100 muty⁻ mutm⁻ E. coli cells transformed with pMAL-MBP-MUTYH plasmid harboring cancer-associated variations. For each variant, 8 colonies were evaluated in triplicate on rifampicin plates (24 total measurements/variants). The 95% confidence limits based on the mean value are shown in the relevant plots. **E** CD spectra of purified cancer-associated MBP-MUTYH. Source data are provided as a Source Data file.

plasmid, the mutation frequency ($f$) was significantly reduced compared to the vector alone (50-fold) demonstrating the activity of the MBP-MUTYH fusion in E. coli. In contrast, transformation with plasmids encoding W103C/W, R241W, R245C, C290W, P295L and C306W CAVs, yielded cultures with high $f$ values (Fig. 3D, Supplementary Table 1). These results are consistent with analysis of the purified proteins that indicated an absence of enzyme activity. In E. coli expressing variants R241Q, R245H, and N238S $f$ values 18-, 23- and 12-fold higher than observed with WT MBP-MUTYH expression; however, these $f$ values are less than the empty vector control, suggesting a potential ability to suppress mutations to some extent. The corresponding mutation suppression assays performed with MUTYH variants V246F and R309C revealed slightly higher $f$ values (5- and 10-fold) than those from cells harboring WT MBP-MUTYH, also suggesting a mildly reduced mutation suppression activity for these two variants despite appearing quite similar to WT in their in vitro activity. Overall, the complementation assays recapitulate the in vitro activity results but provide additional gradations within the variant functional groupings by distinguishing V246F and R309C from WT, and R245H/C from the set of variants that were completely inactive and unable to bind to DNA in vitro.

The impact of MUTYH variations on retention of the metal cofactors and proper folding of MUTYH was assessed via ICP-MS metal analysis and Circular Dichroism (CD) spectroscopy. The MUTYH CAVs that exhibited significant levels of binding to the OG:THF duplex, i.e., N238S, R241Q, V246F and R309C, were found to retain similar levels of Zn and Fe as the WT MBP-MUTYH (Table 1). CD spectra of N238S, R241Q, V246F and R309C featured a strong signal between 200 nm and 240 nm, similar to that of the WT enzyme, consistent with the high alpha helical content of MUTYH and MBP (Fig. 3E)[12]. However, all the other inactive CAVs that that lacked significant DNA binding capacity, exhibited only background levels of both metal ion cofactors (Table 1). In the case of mutations at Trp103 (W103C/R) that is part of a hydrophobic interface of the [4Fe-4S] cluster domain and the catalytic pocket, the CD spectrum indicates a dramatic loss of secondary structure consistent with protein unfolding. Notably, the structural changes caused by Trp103 replacement also impact the secondary structure of MBP based on reduction of CD signal. Destabilizing mutations in passenger proteins that affect MBP structure and vice versa have been reported previously[26,27]. In all other variants that had failed to retain the metal cofactors, there was no significant unfolding

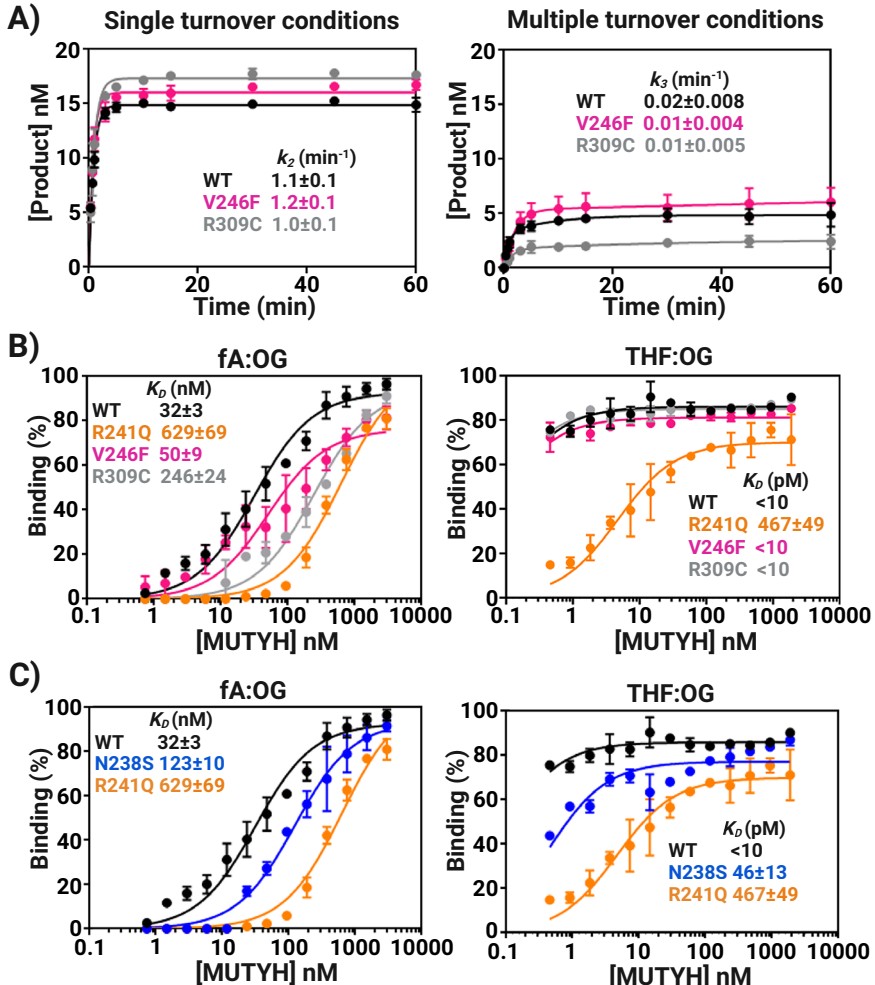

**Fig. 4 | Kinetics of adenine glycosylase activity and dissociation constants, Kd.**
**A** Full-time course adenine glycosylase assays for determination of kinetic parameters for WT, V246F, and R309C MUTYH. Left panel; time course reaction of MBP-MUTYH adenine excision under single turnover condition ([E] = 100 nM active concentration, [A:OG] = 20 nM). Right panel; kinetics of adenine excision under multiple turnover conditions ([E] = 2.5–5 nM active concentration, [A:OG] = 20 nM). Data fitting was used to obtained the glycosylase ($k_2$) and turnover ($k_3$) rate constants (See "Methods" for more information). **B** DNA binding isotherms of electrophoretic mobility shift assays (EMSA) for WT, R241Q, V246F and R309C. Left panel; DNA binding isotherms obtained with substrate adenine analog (fA) across OG (10 pM) titrated with increasing concentrations of MBP-MUTYH (0–190 nM). Right panel; DNA binding isotherms obtained with abasic site analog (THF) across OG (10 pM) titrated with increasing concentrations of MBP-MUTYH (0-3000 nM). **C** DNA binding isotherms obtained for $K_D$ determination with fA:OG- and THF:OG-containing DNA duplex binding with WT, R241Q and N238S MBP-MUTYH. Data for R241Q are shown in both panel B and C for comparison. All experiments were repeated in triplicate and the error bars the standard deviation from the mean. Source data are provided as a Source Data file.

of the protein as indicated by CD spectroscopy. Surprisingly, MUTYH variants R241W, R245H/C, C290W, P295L, and C306W showed a larger signal at the alpha helical region in the CD spectra suggesting an increase in alpha helicity upon loss of the cluster cofactor. Clearly, the CAVs lead to variable impacts on structure that show that the [4Fe-4S] cofactor and surrounding residues are linked to structure and function. In the case of Trp103 substitution, the global structure appears to be destabilized, while in several other examples, such as N238S and R241Q, the structural impact appears to be more local yet nevertheless detrimental for function.

To discern subtleties in functional defects of V246F and R309C MUTYH, we performed detailed adenine glycosylase and EMSA analyses to measure relevant kinetic and binding parameters. Specifically, we measured the base excision step ($k_2$) and product release ($k_3$) rate constants (Scheme 1) with full-time course adenine glycosylase assays of variant enzyme reacting with a 30-bp OG:A duplex under single turnover (STO, [E] > [S]) and multiple turnover (MTO, [E] < [S]) conditions (Supplementary Table 2)[5]. Relative dissociation constants ($K_D$) were measured with the corresponding duplexes replacing the "A"

with either a non-cleavable substrate analog (2′-fluoro-2′-deoxy-yadenosine [fA]) or product (tetrahydrofuran [THF]) via EMSA(Supplementary Table 2)[6,28]. The adenine glycosylase assays (Fig. 4A) revealed that V246F and R309C CAVs had similar base excision $k_2$ and product release $k_3$ rate constants as WT MBP-MUTYH. However, the MTO experiments showed a reduced burst amplitude, indicating a reduced active fraction, for R309C relative to WT and V246F. Both R309C and V246F variants exhibited WT-like high affinity for THF:OG-containing product analog duplex ($K_D$ < 10 pM); notably, these values represent upper limit estimates due to experimental detection limitations (Fig. 4B). The high affinity of MUTYH to the product analog and its observed slow rate of product release is consistent with previous work with MutY enzymes[8]. The R309C variant recognized substrate DNA poorly as assessed by an 8-fold increase in $K_D$ with the fA:OG duplex relative to WT MUTYH. The V246F variant had apparently no detectable impact on substrate recognition as its $K_D$ was comparable to that of WT MUTYH. The low active fraction and increased $K_D$ for R309C suggest a compromised ability to engage the lesion substrate, potentially due to local structural changes at the

external face of the [4Fe-4S] cluster which are transmitted to the DNA-protein interface. These results further illustrate the range of activities that result from CAVs, and the robust activity for V246F and R309C suggest that these variations may erode mutation suppression by other means. For instance, the activity may be reduced due to cellular conditions of high oxidative stress, or by compromised interactions with down-stream repair machinery. Additional studies are warranted to reveal functional defects associated with these CAVs.

Differential impacts of the R241Q and N238S variants on product and substrate DNA affinity were revealed in the measured $K_D$ values with THF:OG and fA:OG-containing DNA duplexes. We observed ~4-fold reduction in the affinity of N238S for the product and substrate analog duplex. In contrast, a larger reduction in affinity for R241Q of 45- and 20-fold for the product or substrate analog, respectively, was observed (Fig. 4C). These reductions in affinity are likely a consequence of removing the electrostatic and H-bonding interactions with the DNA phosphodiester backbone. However, of note, binding defects with R241Q and N238S are not compensated for at the higher concentrations used in the STO glycosylase assays ([E] > $K_D$), as may be expected for non-specific DNA interactions. This suggests that the erosion of binding affinity is substrate specific and reflects an altered binding mode that does not support catalysis. Moreover, unlike CAVs like R241W, which lacks both cofactors and exhibits no glycosylase activity or DNA binding affinity, R241Q and N238S retain levels of Fe and Zn (Table 1) and overall folding (Fig. 3E) like the WT, further supporting distinct alterations caused by these variations that is communicated to the active site to thwart catalysis.

## Impact of R241Q variant on the structural interplay between the [4Fe-4S] cluster and the active site

In order to delineate the structural basis for the unexpected absence of activity for the R241Q MUTYH, we turned to the corresponding variant in *Geobacillus stearothermophilus* MutY (GsMutY), R149Q. Of note, the Arg residue, and its participation in the H-bond network with the [4Fe-4S] cluster is highly conserved in MutY enzymes, and therefore likely plays a similar role (Fig. 2E). However, unlike the mutation in the human homolog, R149Q GsMutY retained measurable adenine glycosylase activity, but with a significant reduction on the base excision rate constant ($k_2$) and turnover rate constant ($k_3$) of ~50- and 5-fold relative to WT GsMutY, respectively (Supplementary Table 3).

Crystals of R149Q GsMutY complexed with a THF:OG-containing duplex diffracted synchrotron radiation to the 1.51-Å resolution limit and the structure was refined through phase extension with simulated annealing, restrained minimization and model rebuilding to yield R/Rfree values of 0.207/0.230 (PDB ID 9BS2). As shown in Fig. 2D, alterations in the GsMutY structure in the immediate neighborhood of position 149 propagate to the [4Fe-4S] cluster and its Cys290 ligand to induce alternate conformations for the metal cofactor in the R149Q variant. Elongated features in the map calculated from anomalous differences clearly define two different conformations of the [4Fe-4S] cluster with a 1.4-Å displacement separating conformation A and conformation B, which refined with approximately equal group occupancies: $q = 0.53$ (A) and 0.47 (B) (Fig. 2D and Supplementary Fig. 9). Another difference apparent for the R149Q GsMutY structure, is a calcium ion chelated by $O^{\delta 1}$ of Asn146. In the WT GsMutY structure, $O^{\delta 1}$ of Asn146 accepts an H-bond from Arg149. Apparently, loss of this Arg149-Asn146 H-bond creates an opportunity for invasion by divalent metal ions, such as $Ca^{2+}$, which is abundantly present in the crystallization condition. Alternate conformations for the [4Fe-4S] cofactor and invasion by $Ca^{2+}$ has been seen previously for a structure of N146S GsMutY CAV (corresponds to N238S in MUTYH) captured with substrate DNA (PDB ID 8DVP) that similarly disrupts the Arg-Asn connection at the Asn[28]. Two structural states for the [4Fe-4S] cofactor, as revealed by these two variants, N146S and R149Q, in proximity to the active site and [4Fe-4S] cluster respectively, suggests that this allosteric network communicates events between the cofactor and active site for critical functional outcomes.

## Molecular dynamics simulations reveal an allosteric network between the [4Fe-4S] cluster and the active site

MD simulations were performed to investigate the structural and dynamic relationships between the [4Fe-4S] cluster and the active site in the WT, R241Q and N238S mutant in both mouse and human homologs, using reported structures (see "Methods"). For simulations based on human MUTYH TSAC, the 1N nucleotide was replaced with an AP site for comparison to the mouse structure. Overall human and mouse structures exhibited similar behavior (Fig. 5, Supplementary Figs. 10–13). The discussion below focuses on the results for the systems for the MUTYH structures, while detailed results for the mMutyh structures are provided in the supporting information (Supplementary Fig. 12 and 13). Analysis on the resulting trajectories were performed including root mean square deviation (RMSD), normal mode analysis (NMA), and energy decomposition analysis (EDA) to understand the dynamic features and differences between the systems (Fig. 5 and Supplementary Figs. 10–13).

NMA indicates that both the WT and R241Q systems show similar behavior in terms of percentage contribution of motion, with the first mode contributing ~90% and the second mode contributing ~10% to the overall motion (Fig. 5, Supplementary Fig. 10), and movie found at Zenodo repository (https://doi.org/10.5281/zenodo.10161357)[29]. However, the first two modes for WT correspond to a breathing-like motion whereas the R241Q system exhibits a rocking-like motion. Conversely, the first mode of the N238S system constitutes about ~69%, while the second mode and third mode contribute ~29% and ~2% respectively to the overall motion. These modes indicate different types of rocking motion. NMA analysis suggests that the mutations change the dynamics of the system to a more rigid rocking motion from breathing motion and specifically, the first normal mode for N238S is drastically reduced with respect to percentage contribution compared to WT and R241Q.

We performed EDA to calculate the non-bonded intermolecular interaction energies (Coulomb and Van der Waals) as a function of specific reference fragments (e.g., residue, nucleotide, etc.). This approach allows us to qualitatively investigate the interactions of individual residues in the allosteric network linking the [4Fe-4S] cluster, the catalytic Asp236 and the AP site. Both Arg241 and Asn238 show significant contributions to the overall stability of the protein with total interaction energy of −423.1 and −66.4 kcal/mol, with respect to residues 241 and 238 respectively (Fig. 5). However, mutation of these residues reduces the overall stability of the systems (−61.9 and −56.8 kcal/mol for R241Q and N238S, respectively). Interestingly, the R241Q mutation severely reduces the energy contributions for the catalytic Asp236 and the AP site, diminishing these from −42.7 and −62.7 kcal/mol for each residue in the WT MUTYH to −4.1 and −2.9 kcal/mol. Furthermore, the difference in non-bonded interaction energy (ΔE) was calculated between the mutant systems and the WT with respect to the [4Fe-4S] cluster. Our results suggest that the mutation of Asn238 to serine results in decreased interactions between several nucleotides/residues and the [4Fe-4S] cluster. Interestingly, for the R241Q variant, the DNA strand containing the AP site shows improved interactions with the [4Fe-4S] cluster (Supplementary Fig. 11), while the other DNA strand is destabilized. In addition, Arg241 is destabilized in WT compared to the R241Q mutant (−49.5 kcal/mol), as both the [4Fe-4S] cluster and Arg are positively charged. The sum of ΔE between N238S and WT is +23.1 kcal/mol and the sum of ΔE between R241Q and WT is +4.9 kcal/mol, which suggests that both mutations result in overall decreased interactions between the [4Fe-4S] cluster, and the rest of the protein compared to WT.

To further investigate the allosteric network between the [4Fe-4S] cluster and the catalytic Asp236 we performed dynamic network

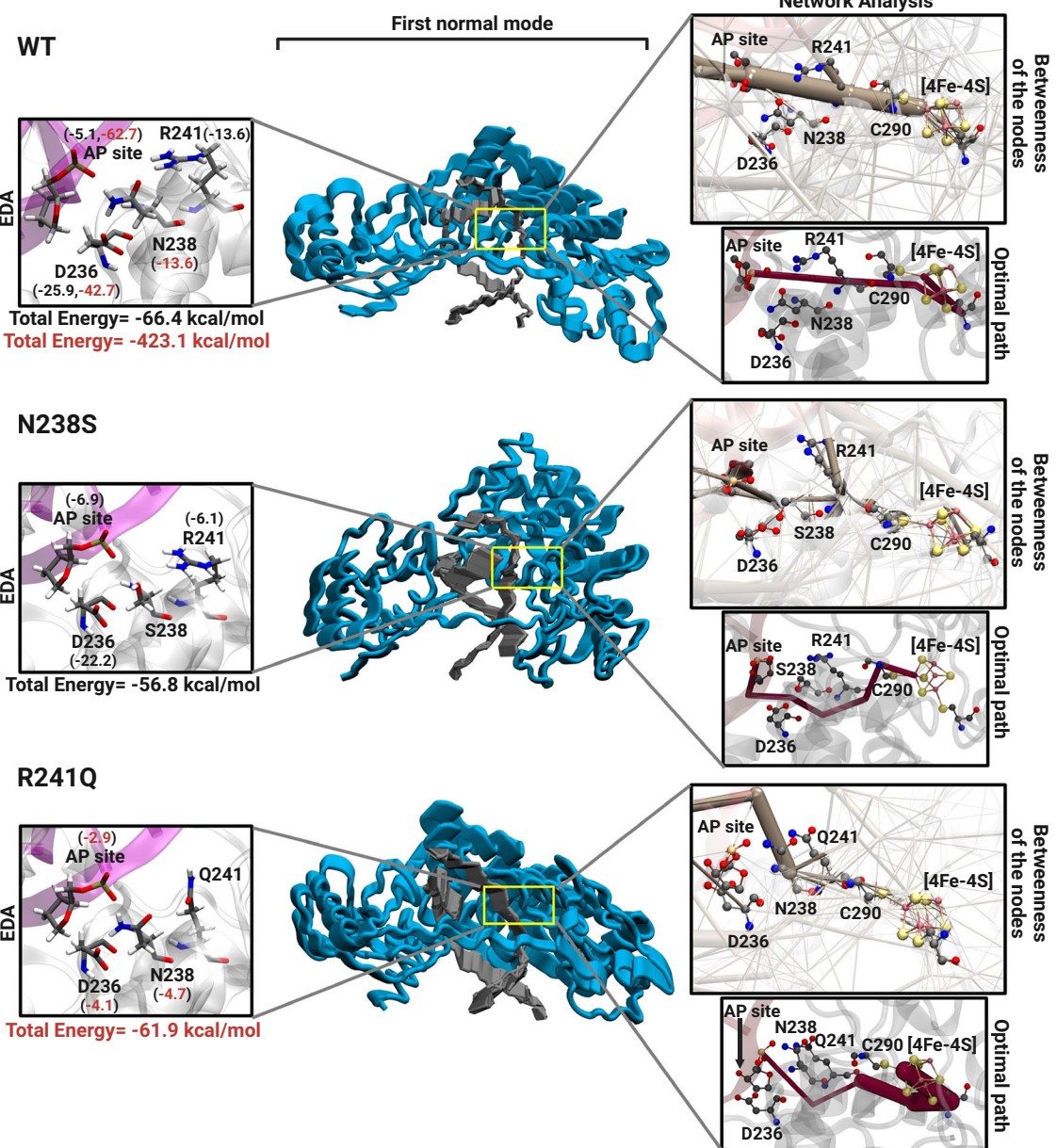

**Fig. 5 | Molecular dynamics simulation.** Energy decomposition (left), normal mode (middle), and network (right) analyses based on the MD trajectories are shown. The left and right panels show a close-up view of the residues that participate in the structural interplay between the [4Fe-4S] cluster and the active site along with values for energy decomposition analysis (EDA, left) and network analysis results (right). The EDA reports the intermolecular non-bonded interactions (Coulomb and van der Waals interactions; kcal/mol) involving Asn238 or Arg241 (black and red values, respectively). Below, the total energy contribution for the system is shown for Asn238 (black) and Arg241 (red) and corresponding cancer-associated mutants. For the network analysis the betweenness of the nodes involved in the multi-motif bridge is shown in brown (right-upper panel) and the optimal path between the AP site and [4Fe-4S] cluster is displayed in magenta (right-lower panel). The first normal mode is shown in the middle where the MUTYH and DNA are shown in blue and gray, respectively.

analysis. The nodes connecting the AP site and Arg241 in the WT structure show a significant betweenness. This is disrupted by both mutations, N238S and R241Q (Fig. 5). Additionally, the betweenness of the AP site and Asn238 shows a connection associating these two nodes in the WT system, which is broken in both mutant systems. The optimal path connecting the [4Fe-4S] cluster to the AP site was determined for all systems. The optimal path between the AP site and the [4Fe-4S] cluster in WT is through Arg241 (Fig. 5). However, this optimal path is altered for N238S, in which Cys290 is involved in a series of the H-bonds that connect to the [4Fe-4S] cluster. Furthermore, this optimal path is completely changed for the R241Q system and an alternative path through the protein is found without involving the residues in the H-bond bridge. Altogether, analysis of the MD

simulations predicts the existence of a network connecting the [4Fe-4S] metal site to the catalytically critical Asp236. We suggest that this network serves as an allosteric regulator to ensure adenine excision from rare OG:A lesions but not highly abundant T:A bps, and as a means to control enzyme function in response to cellular conditions.

## Discussion

Herein, we reported the first human MUTYH transition state analog complex (MUTYH TSAC) structure and delineated structural and functional consequences of 12 CAVs near to the [4Fe-4S] cluster of MUTYH. These results lead to a significant conceptual advance in our understanding of this critical base excision DNA repair enzyme. Whereas we and others have speculated as to the nature of a functional connection

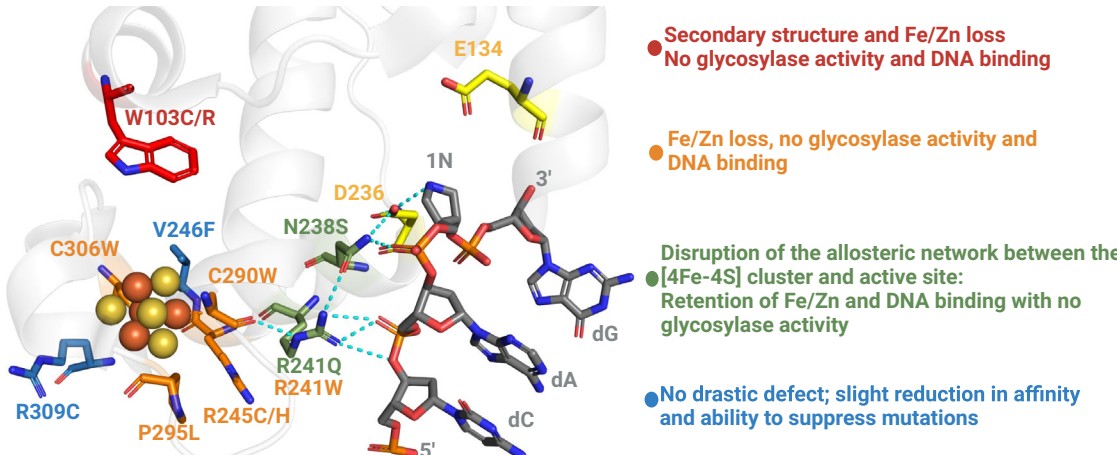

**Fig. 6 | Summary of functional impacts of MUTYH [4Fe-4S] CAVs.** The different functional impacts of MUTYH cancer-associated variants in proximity to the [4Fe-4S] cluster allowed for grouping into distinct classes. The impact statements correlate with the color of the residue where the variants are localized.

between the [4Fe-4S] metal cluster and the active site[18,28], herein, we define how that connection is mechanistically established. The MUTYH TSAC structure provided a means to correlate position and nature of substitution of the CAVs with the impact on MUTYH activity; an annotated structure-activity summary of the 12 variants studied herein is shown in Fig. 6. Our results with MBP-MUTYH, consistent with previous work with *E. coli* MutY, show that the loss of the [4Fe-4S] cofactor, either by denaturation, chelation or by mutation of coordinating ligands or surrounding residues, leads to loss of glycosylase activity and DNA binding capabilities without drastic changes in secondary structure[4,14,15,18,30]. Notable exceptions are MUTYH CAVs W103C and W103R which disrupt the hydrophobic packing between the cluster domain and the active site, highlighting the close association of these domains. In previous work, McDonnell et al. identified and characterized the biochemical and redox properties of C306W as an MBP-MUTYH recombinant protein[18]. These authors found similar results to those observed herein with C306W: compromised glycosylase activity and no detectable DNA binding. However, McDonnell et al. reported that C306W MBP-MUTYH retained a low level of [4Fe-4S] ((9%) based on [Fe]), that was sensitive to oxidative degradation. In addition, Komine et al. reported similar results for mutation suppression activity of W103R, R241W, R245C/H, V246F, C290W, and P296L in a MutY-deficient *E. coli* strain[24]. Importantly, our results and these studies underscore the functional importance of the [4Fe-4S] cofactor in MUTYH.

Our inspection of the human MUTYH structure revealed an intricate H-bonding network which spans from the [4Fe-4S] cluster, Cys290 ligand, Arg291, Asn238 up to the catalytic Asp236 and the transition state analog 1N. Comparison with mouse MUTYH and bacterial GsMutY structures shows this H-bonding network to be conserved in structure across evolution (Fig. 2E). Curiously, the network includes two CAVs, R241Q and N238S, that each displayed WT levels of Fe and Zn ion and were capable of binding to substrate and product DNA but were completely inactive as glycosylases. The structures of MUTYH, along with those of GsMutY Arg→Gln and Asn→Ser variant[28], and the MD simulations illuminate how the allosteric network between the [4Fe-4S] cofactor DNA binding site and the active site regulates positioning and protonation state of the catalytic Asp required for base excision catalysis.

In MD/QM studies previously reported for WT and the Asn→Ser variant of GsMutY[31,32], the Asn-Asp interaction was found to be quite dynamic during catalysis. Indeed, in the bacterial enzyme MD simulations, the H-bond that connects Asn to Asp breaks halfway through catalysis, with a concomitant change in Asp protonation. This is consistent with the increase in pKa of the Asp we observed previously with

N146S GsMutY relative to the WT enzyme[28]. Notably, we also observed an altered position of the Purine base in the active site in the structure of N146S GsMutY with an OG:Purine substrate that likely hinders protonation of N7 by Glu43. These studies combined with those herein illustrate the exquisite control exerted over the base excision chemistry of MutY enzymes to ensure DNA repair fidelity.

Sequence and structural analyses of other [4Fe-4S] cluster-containing Helix-hairpin-Helix (HhH) DNA glycosylases reveal a similarly conserved allosteric network connecting the [4Fe-4S] cluster and the active site (Fig. 2E). The thymine-DNA glycosylase MIG has an identical, Cys-Arg-Asn-Asp network[33]. The EndoIII/NTHL1 glycosylase conserves a similar H-bond connectivity with His instead of Asn (Cys-Arg-His-Asp; Fig. 2E)[34] The mechanistic details of other HhH [4Fe-4S] cluster containing BER glycosylases have not been studied as extensively as MutY, though likely share an $S_N1$-like mechanism[7] (Fig. 7). Despite the differences in substrate processed by these other HhH DNA glycosylases, it is quite striking that there is a high degree of conservation of the residues to maintain the H-bond network between the [4Fe-4S] cluster and the catalytic pocket. This suggests that such a multi residue-bridging motif has been a functionally important structural element throughout the evolution of [4Fe-4S] cluster-containing HhH BER glycosylases. Similarly, in single domain plant-type ferredoxins, allostery between a loop and a [2Fe-2S] cluster 20 Å apart has been shown to involve minimal structural perturbations propagated by short-range interactions and display concordant patterns of evolution[35,36].

The impact on catalysis of mutations to the Arg-Asn residues linking the [4Fe-4S] cluster to the active site suggests a means by which changes in the redox state of the [4Fe-4S]$^{2+/3+}$ cluster alters base excision catalysis. In previous collaborative studies with the Barton laboratory, the [4Fe-4S]$^{2+}$ cluster in *E. coli* MutY was shown to become redox active upon DNA binding, facilitating oxidation from [4Fe-4S]$^{2+}$ to [4Fe-4S]$^{3+}$[17,37–39]. In addition, a variety of DNA repair and replication enzymes harboring a [4Fe-4S] were shown to have DNA-mediated dependent redox activity[40]. Notably, in the case of EndoIII, oxidation was shown to increase DNA affinity[41]. Based on these studies, Barton and co-workers have proposed that [4Fe-4S] cluster nucleic acid processing enzymes utilize the metal cofactor to sense the DNA integrity and locate DNA lesions by DNA association versus dissociation controlled by the cluster redox state and communicated via DNA-mediated charge-transport[37,40,42]. In the work herein, many of the [4Fe-4S] CAVs were found to be destabilizing to the [4Fe-4S] cluster; this may be a reflection of an altered redox potential. In contrast, the V246F and R309C variants exhibited WT-like enzyme activity,

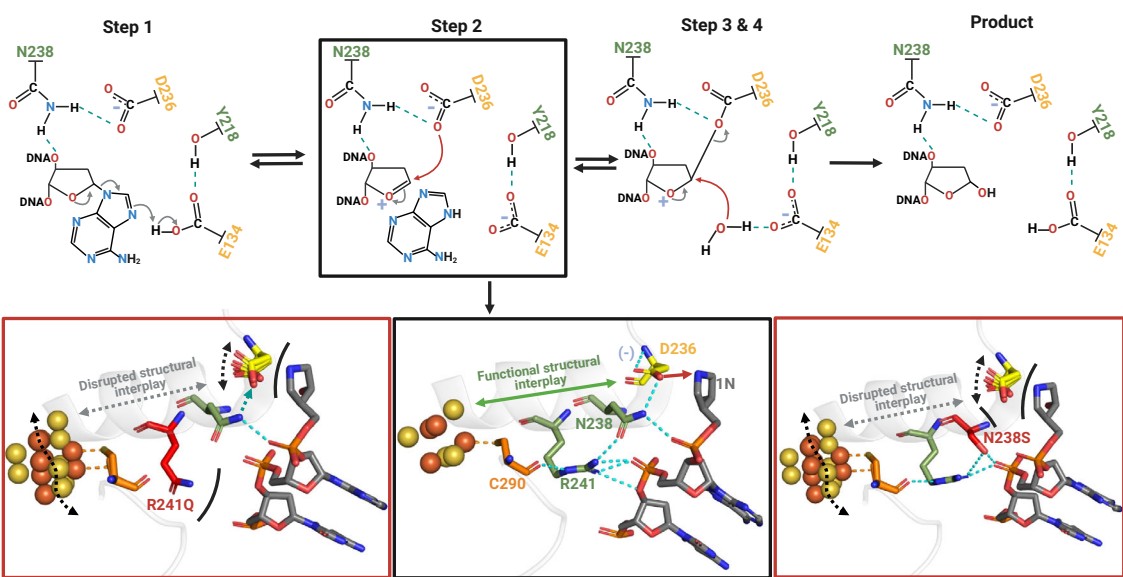

**Fig. 7 | Proposed catalytic mechanism for MUTYH and disrupted mechanism in R241Q and N238S cancer-associated variants.** The adenine excision catalyzed by MUTYH is initiated with the protonation of the adenine by Glu134 which is followed by the glycosidic bond cleavage and formation of an oxacarbenium ion intermediate. Asp236 is proposed to attack the oxacarbenium ion at C1' to stabilize it by a covalent intermediate. Finally, Glu134 activates a water molecule for nucleophilic attack at C1 to form the AP site product. The structural interplay between the [4Fe-4S] cluster is important for step 2 of catalysis, where Asp236 attacks the oxacarbenium ion. Asp236 has to be oriented near the N-glycosidic bond and deprotonated for the nucleophilic attack. Such orientation and deprotonation status of Asp236 is

modulated by the H-bond with Gln238 and allosterically through the [4Fe-4S] cluster via the Cys290-Arg241-Asn238-Asp236 structural bridge. Upon R241Q and N238S mutations the H-bond between Asp236 and Asn238 becomes unstable (cyan dashed arrow) or is broken (black curved lines). With the structural bridge thus compromised the position, protonation state, and charge of catalytic Asp236 position are also impacted (double-headed dashed arrows) with the net consequence of impeding nucleophilic attack of the oxacarbenium ion. This pathological scenario is accompanied by structural instability of the [4Fe-4S] cluster and loss of DNA-MUTYH interactions.

however, the cluster in these enzyme variants may exhibit reduced activity due to cluster instability when challenged by conditions of oxidative stress. Two computational studies using QM/MD calculations and hole hopping analyses, have defined a charge-transport pathway between the [4Fe-4S] cluster and the active site (OG or A) using structures of GsMutY[43,44]. Indeed, these studies have suggested that Arg241[43] and Trp103[44] are part of the charge transport pathway and that their mutation would hamper this redox communication. Although many details of the cofactor redox roles in MUTYH remain to be elucidated, it is clear that the [4Fe-4S] cluster plays multiple essential roles in the enzyme's function.

The current study provides functional and structural information on MUTYH CAVs and reveals fundamental features that aid in understanding cancer etiology at an atomic level. Through studying MUTYH CAVs, we provide structural and biochemical evidence of an allosteric role for the [4Fe-4S] cluster in HhH DNA glycosylases, where an H-bond network coordinates DNA binding near the [4Fe-4S] cluster to adenine excision via the key catalytic Asp. CAVs that disrupt the allosteric network, such R241Q and N238S, compromise MUTYH function, providing for increased mutagenesis to trigger carcinogenesis. The influence of changes near the [4Fe-4S] cluster and the active site, also suggests a potential regulatory role in DNA repair where DNA interactions at the [4Fe-4S] cluster binding domain are transmitted to the active site to reduce or enhance base excision (Fig. 5). Alterations at the [4Fe-4S] may be influenced by the DNA context and or redox status of the cell. Under conditions of high oxidative stress, the loss of the [4Fe-4S] cluster may be a means to down-regulate MUTYH activity to prevent accumulation of unrepaired AP sites that may lead to genotoxic strand breaks as evidenced in telomere instability[45]. In contrast, under conditions of slightly elevated oxidative stress, cluster oxidation may be a way to augment MUTYH binding and enhance its activity. Of note, structurally, the [4Fe-4S] cluster is not an inert element. The oxidized state of the cofactor has an increased net positive charge (+3 versus +2)

and smaller Van der Waals volume and surface area than the reduced cofactor (~550 Å$^3$ and ~504 Å$^2$ versus 570 Å$^3$ and 530 Å$^2$, respectively)[46]. These changes in cofactor size and charge may be transmitted to the active site via the newly identified H-bond network to modulate MUTYH activity. Thus, we propose that the characteristic 2+ redox state of the cofactor of the DNA-free MUTYH represents the less active allosteric mode where the reduced [4Fe-4S]$^{2+}$ cofactor positions the catalytic Asp in an orientation and protonation state that are suboptimal to perform catalysis (Fig. 7). However, upon DNA binding and oxidation to the [4Fe-4S]$^{3+}$ state cluster, adenine excision may be enhanced via control of the catalytic Asp, representing a more active allosteric state. Interestingly, under oxidative stress MUTYH along with OGG1 activities are reported to exert telomeric instability by means of replicative stress[45,47]. Therefore, this type of allostery may be a mechanism to control aberrant MUTYH activity under oxidative stress.

The sensitivity of MUTYH activity to alterations and loss of the cluster also suggest this as a site for targeting with small molecules. Indeed, TEMPOL, a stable nitroxide, was recently shown to cause oxidative degradation of the [4Fe-4S]$^{2+}$ cluster in the RNA-dependent RNA polymerase of SARS-CoV-2 to block viral replication in animals[48]. Remarkably, optimal function of MUTYH and likely other HhH glycosylases depends on its [4Fe-4S]$^{2+}$ cluster, however the reliance on a sensitive redox cofactor provides avenues for its degradation under oxidative stress, which would explain how chronic inflammation and associated oxidative stress contributes to cancer etiology. Ironically, the "Achilles heel" of MUTYH may be its [4Fe-4S] cluster that could be targeted in cancer cells that have become reliant on MUTYH to survive.

## Methods
### Human *MUTYH* cloning and mutagenesis
A codon optimized human *MUTYH* gene (beta 3 isoform) for *E. coli* overexpression was designed and purchased from IDT as a gBlock (See Supplementary information for ORF's sequence). The *MUTYH* ORF is

devoid of the initial fourteen codons to alleviate protein toxicity as previously suggested[18]. The *MUTYH* gBlock was subcloned into pJET2.1 vector and ultimately cloned into a modified version of pET28a vector using NdeI and NcoI restriction sites. The modified version of pET28 allows the overexpression of MBP-MUTYH protein with 2 histidine tags both at N- and C-terminus regions. Two internal TEV protease cleavage sites were introduced to remove the His-tags and MBP segments from the MUTYH protein for crystallography. A scheme of the final pET28-MBP-MUTYH design is included in supplementary data (Supplementary Fig. 2). For crystallography the human *MUTYH* gene was trimmed, removing the initial 34 and last 28 codons, analogous to N- and C-terminal truncations found in the murine structure[13].

Mutagenesis of the pET28-MBP-MUTYH construct was carried out by PCR-driven overlap extension[49]. Due to toxicity of pET28-MBP-MUTYH plasmid in GT100 *muty⁻ mutm⁻ E. coli* strain, for complementation experiments we used the construct pMAL-MBP-MUTYH which contains a non-codon-optimized *MUTYH* gene as previously reported[25]. The mutagenesis of pMAL-MBP-MUTYH construct was done using the Q5 site-directed mutagenesis kit from New England BioLabs (Catalog no. E0554S).

## MUTYH overexpression and purification

The overexpression and purification of MUTYH was carried out as previously reported[50]. Briefly, a BL21(DE3) strain containing pRKISC and pKJE7 vectors was used as the expression host. The pRKISC plasmid co-expresses the [4Fe-4S] cluster assembly machinery[12], and pKJE7 coexpresses dnaK, dnaJ and grgE chaperones[51]. The BL21(+pKRISC +pKJE7) strain was transformed with the pET28-MBP-MUTYH construct and plated onto Luria Broth plates supplemented with 50 μg/mL of kanamycin (Kan), 15 μg/ mL of tetracycline (Tet) and 34 μg/mL of chloramphenicol (Cam). Colonies obtained from the transformation were used to inoculate 2 L of Terrific broth media supplemented with Kan+Tet+Cam in a 4 L flask with a flat bottom and grown for 6 h at 37 °C with shaking at 180 rpm until an $OD_{600nm}$ of at least 1.5. The culture was cooled without shaking 1 h at 4 °C, and then protein production was induced with IPTG (0.25 mM) and addition of ferrous sulfate (0.1 g) and ferric citrate (0.1 g). The overexpression was carried out at 15 °C for 16–24 h with continuous shaking. Bacteria pellets were obtained by centrifugation (6000 × *g*/10 min/4°) and stored at −80 °C until needed.

For protein purification the pellets were thawed and resuspended in Lysis buffer (30 mM Tris [pH 7.5], 1 M NaCl, 30 mM 2-mercaptoethanol and 10% glycerol) supplemented with 1 mM of phenylmethylsulfonyl fluoride. The cellular lysis was carried out by sonication on ice in 20 s cycles using a Branson Sonifier 250 followed by centrifugation at 15 000 × *g* for 50 min at 4 °C. The clarified supernatant was incubated with 1.5 mL of Ni²⁺ NTA resin (Qiagen; catalog no. 30210) for 1 h at 4 °C with rotation. The slurry was poured over a PD10 column (Cytiva; catalog no. 17-0435-01) and allowed to flow through via gravity. The protein-loaded resin was washed with at least 25 mL of Lysis buffer followed 10 mL of elution buffer (30 mM Tris [pH 7.5], 200 mM NaCl, 30 mM 2-mercaptoethanol, 10% glycerol and 500 mM imidazole). Removal of imidazole immediately following elution proved to be critical for protein stability. To accomplish buffer exchange, the protein was subjected to heparin-affinity chromatography where the elution was diluted with an equal volume of buffer A (30 mM Tris [pH 7.5], 10 % glycerol, 1 mM DTT and 1 mM EDTA) to reduce the NaCl concertation to 100 mM. The diluted nickel elution was loaded onto a 1 mL Heparin column (Cytiva; catalog no. 17040601) previously equilibrated with 10% buffer B (30 mM Tris [pH 7.5], 1 M NaCl, 10 % glycerol, 1 mM DTT and 1 mM EDTA). The heparin column containing the bound protein was washed with 25 mL of 10% buffer B and the protein was eluted using linear gradient of buffer A and B (10-100%) over 45 min with a flowrate of 1.5 mL/min using an AKTA FPLC instrument (GE Healthcare). The fractions containing pure MBP-

MUTYH protein were analyzed by SDS-PAGE (Supplementary Fig. 6) and concentrated using Amicon ultracentrifugation filters (30,000 MWCO; catalog no. UFC5030). The protein concentration was estimated by measuring the 280-nm UV absorbance with an extinction coefficient of 152,470 M⁻¹ cm⁻¹. The purified protein was then aliquoted and stored at −80 °C.

The protein for crystallography was treated with TEV protease after nickel-affinity chromatography with a ratio 30:1 (w/w) of MUTYH:TEV at 4 °C for 16 h in 10% buffer B to remove the His-tag and MBP tags. A second Heparin affinity chromatography step using a 5 mL Heparin column was performed to remove the released tags, followed by size-exclusion chromatography using Superdex 200 column with 20% buffer B. Purification of R149Q GsMutY protein was carried out as previously reported for the N146S *Gs* MutY enzyme[28].

## Preparation of oligonucleotide substrates

The 1N transition state analog (3R,4R)-3-(hydroxymethyl) pyrrolidine-1-ium phosphoramidite was synthesized with modifications of literature procedures[52], as described in the Supplementary Information. The 1N, OG, FA, and A containing 2′deoxyribonucleotides used for crystallography, binding, and kinetic experiments were synthesized at the University of Utah DNA and peptide synthesis core facility. The OG-containing DNA strands were cleaved from the column and deprotected using ammonium hydroxide with 0.25 M 2-mercaptoethanol for 17 h at 55 °C. All the oligonucleotides were HPLC-purified, desalted with Sep-Pak C18 desalting cartridge (Waters) and correct mass confirmed by matrix-assisted laser-desorption/ionization (MALDI) mass spectrometry at the UC Davis Campus Mass Spectrometry Facility (cmsf.ucdavis.edu). The oligonucleotides sequences used in this study are listed in supplementary table 7.

## Crystallography

For human MUTYH crystallization, conditions were optimized from those previously reported with the murine enzyme[13]. The 1N- and OG-containing oligonucleotides used for crystallography (Duplex 1; Supplementary Table 7) were annealed to 1:1 ratio (263 μM) in 30 mM Tris [pH 7.5] by heating at 90 °C for 5 min followed by slow cool annealing to 4 °C. The 1N:OG duplex I (263 μM) was added to MUTYH protein (236 μM) in buffer containing 30 mM Tris [pH 7.5], 100 mM NaCl and 0.5 mM DTT. The resulting MUTYH-DNA complex (118 μM) was incubated for 20 min at room temperature and mixed with crystallization solutions in 1:1 ratio in 3 μL final volume onto a coverslip (Hampton Research). The best quality crystals in terms of size and X-ray diffraction resolution limit were obtained in 0.1 M Bis-Tris [pH 5.5], 0.2 M ammonium sulfate and 20% PEG 3350. The crystals were grown at room temperature by the hanging-drop vapor-diffusion method. Golden rod or needle-like crystals grew over a 12–24 h window. Crystals were harvested directly from the drop and flash cooled in liquid nitrogen. X-ray diffraction data were collected with 0.2° oscillation on beamline 24-ID-E at the Advanced Photon Source (Argonne National Laboratory). Crystallization and X-ray diffraction experiment of R149Q GsMutY and the product analog (THF):OG duplex were carried out using methods similar to those reported previously with N146S and WT GsMutY with DNA[8,28]. Briefly, clusters of crystals containing R149Q GsMutY in complex with THF:OG DNA duplex I were grown with 350 μM of DNA-protein complex in 14% PEG, 400 mM Ca(OAc)₂ and 2% ethylene glycol (pH 8.5). The final individual crystals feasible for X-ray diffraction experiments were grown by microseeding with 10 X dilution of crushed crystals clusters initially obtained.

Diffraction data for the MUTYH-DNA complex structure were processed with XDS and scaled with XSCALE[53]. The mouse Mutyh-DNA complex crystal structure (PDB ID: 7EF9) was used as the search model to obtain initial phases by molecular replacement with Phaser as implemented by PHENIX[54] and refined with iterative cycles of torsion angle simulated annealing, restrained minimization, and manual

model building to yield final R/Rfree values of 0.180/0.209. The structure was refined using PHENIX including 1N, OG, and [4Fe-4S] cluster coordination restraints[54]. The statistics in data processing and model refinement are shown in Supplementary Table 4. The asymmetric unit includes two copies of the hMUTYH-DNA complex. X-ray diffraction data processing, and molecular replacement and refinements for R149Q GsMutY were carried out as previously reported for the N126S GsMutY variant[28] using GsMutY-TSAC structure (PDB ID: 6U7T)[8] as model to obtain the initial phases (Supplementary Table 4). Calcium ions were placed as indicated by coordination geometry analysis provided by the validation tool "highly coordinated waters…" within Coot[55]. Although data were measured far from the optimal wavelength, calcium ions were in clear 4-sigma anomalous difference peaks. All figures depicting structure were generated using PyMOL (The PyMOL Molecular Graphics System, Version 2.0 Schrödinger, LLC), UCSF Chimera[56], and Coot[55]. Coordinates for the human MUTYH-transition state analog complex and the GsMutY R149Q bound with DNA containing THF have been deposited in the Protein Data Bank with PDB IDs 8FAY and 9BS2.

### Coevolutionary analysis

A multiple sequence alignment including 687 amino acid sequence of Archaea, Bacteria, and Eukaryote MutY homologs was generated using the MUSCLE algorithm[57] as implemented with the Geneious software package (Version 4.8, Biomatters). The MSA is included in the SI files. The MUSCLE-generated MSA was manually curated and uploaded into the MISTIC web server[58] along with coordinates for chain A of the human MUTYH structure. The coevolutionary analysis was run with the default parameters and results were visualized using the tools incorporated within the MISTIC web server.

### Inductively Coupled Plasma-Mass Spectrometry (ICP-MS)

Metal analysis by ICP-MS used purified WT and mutant MBP-fusion and MBP free MUTYH proteins that was buffer exchanged to 20 mM Tris-HCl [pH 7.5], 250 NaCl and 10 % glycerol. Samples and blanks were prepared as previously described[3] in a range of 500–250 μL and submitted in triplicate to the UC Davis Interdisciplinary Center for Inductively Coupled Plasma-Mass Spectrometry (https://icpms.ucdavis.edu).

### Glycosylase assay and binding experiments

The glycosylase activity of MUTYH was measured under single turnover and multiple turnover conditions (STO and MTO) to determine the base excision rate ($k_2$) and turnover ($k_3$), respectively as previously reported[5,12] following a minimal kinetic scheme as described below.

$$MUTYH + (DNA)s \underset{k_{-1}}{\overset{k_1}{\leftrightarrow}} MUTYH \cdot (DNA)s \overset{k_2}{\rightarrow} MUTYH \cdot (DNA)p \overset{k_3}{\rightarrow} MUTYH + (DNA)p$$

$$(1)$$

To measure binding affinity of the MAP variants we utilized Electrophoretic Mobility Shift Assay (EMSA) with uncleavable fluorinated-adenine (fA):OG-containing DNA duplex to determine $K_D$ for substrate and tetrahydrofuran (THF):OG-containing DNA duplex for product affinity, as previously reported[6,28]. Briefly, the 5′-end of radiolabeled 2′-FA or THF-containing strand was annealed to its complementary strand with OG and a DNA master mix was prepared in buffer containing 40 mM Tris pH 7.6, 2 mM EDTA, 200 M sodium chloride, 20% (w/v) glycerol, 0.2 mg/ml BSA, 2 mM DTT and 20 pM of radiolabeled Duplex 2 (Supplementary table 7). Equal volumes of DNA master mix were combined with enzyme in decreasing concentrations (prepared at 4 °C in dilution buffer containing 20 mM Tris pH 7.6, 10 mM EDTA and 20% glycerol) and incubated for 30 min at 25 °C. The enzyme bound DNA was separated from the unbound DNA using 6%

nondenaturing polyacrylamide gel ran with 0.5X TBE buffer at 120 V for 2 h at 4 °C. The gels were dried and quantified, and the data was fit to single binding isotherm model to derive the dissociation constant, $K_D$ adjusting the data to nonlinear regression fit of one site-specific binding as described in Eq. 2.

$$Y = \frac{B \max X}{K_D X} \qquad (2)$$

Where X is the concentration of MUTYH, Y is the specific binding (%), Bmax is the maximum binding. Kinetics and EMSA experiments were carried out in triplicate.

### Circular dichroism spectroscopy

Circular dichroism spectroscopy was performed with 0.1 mg/mL of protein in 30 mM Tris [pH 7.5] and 50 mM sodium sulfate, using a 1 mm CD quartz cuvette at room temperature and the Jasco J720 CD spectrophotometer, scanning at a range of 190–240 nm. The data was acquired by averaging triplicate scans and normalized to millidegree to delta epsilon ($\Delta\varepsilon$, $M^{-1}\,cm^{-1}$).

### Rifampicin resistance assay

MBP-MUTYH-pMAL construct was used to analyze mutation suppression activity of CAVs in *E. coli* as previously reported[12]. Briefly, we measured the mutation frequency (as determined by RifR colonies) of GT100 *muty⁻ mutm⁻ E. coli* strain transformed with the WT MBP-MUTYH-pMAL construct or corresponding CAVs relative to the parent strain. For each variant, 8 colonies were evaluated in triplicate on rifampicin plates (24 total measurements/variants), and analyzed as described previously[59]. The estimation of the mutation frequency (*f*) was carried out as follows.

$$f = \frac{median\ number\ of\ resistant\ colonies}{average\ number\ of\ vaible\ colonies} \qquad (3)$$

### Molecular dynamics simulations

The crystal structure for the human MUTYH-DNA complex reported herein was modified to include missing regions as modeled with SWISS-MODEL server[60]. The crystal structure of mouse MUTYH-DNA complex with the PDB ID 7EF8 [13] was used as the initial model. A comparative protein structure modeling was performed, and the missing regions were incorporated with MODELLER 10.4[61]. The mutations of both human and mouse models were introduced using UCSF Chimera[56]. Three systems were considered for each MUTYH and mMutyh models, human: wild type (WT), N238S, R241Q and mouse: WT, N209S, and R212Q. 8-oxo-7,8-dihydro-guanine (OG), AP site, [4Fe-4S] cluster and $Zn^{2+}$ Linchpin motif needed to be parameterized prior to the MD simulation. The azaribose (1N) transition state in the MUTYH-TSAC structure was converted to an AP site using UCSF Chimera. The AMBER force field parameters for OG and AP site were obtained from the AMBER parameter database[62] and the missing parameters were calculated by ANTECHAMBER[63,64]. The parameters of the [4Fe-4S] cluster were obtained from the publication by Squier and co-workers[65] and the parameters of the $Zn^{2+}$ Linchpin motif were obtained from the Zinc AMBER force field[66]. Side chain clashes and protonation states were assessed using ProPKA[67] and MolProbity[68]. All systems were prepared with the Leap module[69] of AMBER21[70] by solvation in a TIP3P[71] cubic box extending a minimum of 10 Å distance from the edge of the protein, and neutralized while setting the ionic strength to 50 mM KCl (Supplementary Table 6).

All molecular dynamics (MD) simulations were carried out with the AMBER ff19SB[72], gaff[63], and OL15[73] force fields with the AMBER21 pmemd.cuda program[70]. Initially, protein and DNA were minimized with a restraint of 300 kcal mol⁻¹ Å² for 500 cycles using conjugate

gradient, continuing for 6000 cycles of steepest descent. Subsequently, each system was heated to 300 K gradually by 50 K in 10,000 MD step intervals at constant volume using Langevin dynamics[74], with the protein restrained by a force constant of 500 kcal mol$^{-1}$ Å$^2$. Next, the systems were equilibrated via gradually reducing the restraints on the protein and DNA using the NVT ensemble with a 1 fs time step (Supplementary Table 5). The production simulations were carried in triplicate for 500 ns each using the NPT ensemble, with a 2 fs time step, without restraints. Constant temperature (300 K) and pressure (1.0 bar) were maintained using the Langevin thermostat and Berendsen barostat[75,76]. All bonds involving hydrogen atoms were constrained using SHAKE[77]. Long-range electrostatic interactions were addressed with the Smooth Particle-Mesh Ewald method, while Van der Waals interactions were controlled by employing the Isotropic Periodic Sum method with a real-space distance of 10 Å[78,79].

The AMBER21 CPPTRAJ[80] module was used to calculate the RMSD and root-mean-square fluctuation in the production trajectories. NMA was carried out using ProDy[81]. EDA was performed using Fortran90 program to calculate intermolecular non-bonded interactions (Coulomb and Vander Waals interactions) between a residue of interest and the rest of the system[82,83]. Dynamic network analysis was conducted using the Dynamic Network Analysis Python package[84,85]. Network analysis was performed to investigate the node-node interactions in all systems. In this analysis, each residue is represented by nodes, where each amino acid is depicted by a single node on the alpha-carbons and each nucleotide by two nodes on the backbone phosphorous and nitrogen atom in the nitrogenous base. The shortest distance between two nodes is calculated to identify which nodes are in contact. If a pair of nodes maintain contact in more than 75% of the simulation frames with a distance ≤4.5 Å, they are considered to be in contact and those nodes are connected by edges. The weight of an edge between nodes corresponds to the probability of information transfer along that connection as calculated based on the correlation values of two residues. The count of shortest paths that pass through an edge in the network are described as the betweenness of an edge. This serves as a measure for assessing the significance of the edge for communication within the network. Furthermore, the optimal path which refers to the most efficient or shortest path was calculated between AP site node and Fe-S cluster node using correlations as weights to determine the shortest distances between these two nodes.

### Reporting summary

Further information on research design is available in the Nature Portfolio Reporting Summary linked to this article.

## Data availability

The atomic coordinates and structure factors generated in this study have been deposited in the Protein Data Bank (www.rcsb.org) under accession codes 8FAY (human MUTYH complexed with DNA containing the transition state analog 1N) and 9BS2 (Gs MutY R149Q complexed with DNA containing the product analog THF). The initial coordinates and parameter files for each WT and mutant systems in human and mouse MUTYH generated as part of the molecular dynamics studies, as well as parameters describing the [4Fe-4S] cluster and Zn$^{2+}$ Linchpin motif have been deposited to the Zenodo repository[29] found at https://doi.org/10.5281/zenodo.10161357. Source data are provided with this paper.

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

## Acknowledgements

We thank Razan Kaddoura for technical help with mutagenesis and Savannah Conlon for purification of OG oligonucleotide for duplex 2. We also thank Madhu Budamagunta and John Voss for access and experimental assistance to obtain the CD spectra. C.H.T.A. was supported in part by a postdoctoral fellowship from UC-MEXUS/CONACYT and N.T. was supported by an AGEP-Graduate Research Supplement (CHE-2039752). This work was supported by research grants from the National Cancer Institute (CA069785 to S.S.D.) and the National Institute of General Medical Sciences (GM108583 and GM151951 to G.A.C.), and the National Science Foundation (CHE:CLP- 1905249, 2204229 to M.P.H.). In addition, computing time from University of North Texas CASCaM supported by the National Science Foundation (OAC-2117247 to G.A.C.) and from the University of Texas at Dallas Cyberinfrastructure are gratefully acknowledged. Part of this work is based upon research conducted at the Northeastern Collaborative Access Team beamlines, which are funded by the National Institute of General Medical Sciences from the National Institutes of Health (P30 GM124165). The Eiger 16M detector on the 24-ID-E beam line is funded by a NIH-ORIP HEI grant (S10OD021527). This research used resources of the Advanced Photon Source, a U.S. Department of Energy (DOE) Office of Science User Facility operated for the DOE Office of Science by Argonne National Laboratory under Contract No. DE-AC02-06CH11357. We thank ALS Beamline 5.0.1 staff Marc Allaire and Core Ralston for assistance with data collection at the Advanced Light Source. The Berkeley Center for Structural Biology is supported by the Howard Hughes Medical Institute, Participating Research Team members, and the National Institutes of Health, National Institute of General Medical Sciences, ALS-ENABLE grant P30 GM124169. The Advanced Light Source is a Department of Energy Office of Science User Facility under Contract No. DE-AC02-05CH11231. The Pilatus detector on beamline 5.0.1 was funded under NIH grant S10OD026941.

## Author contributions

C.H.T.A and S.S.D conceived the project; C.H.T.A., U.C.D., S.S.D. and G.A.C designed the experiments; C.H.T.A., N.T., M.H., M.D., N.H.G., performed biochemical and structural experiments; U.C.D., C.H.T.A., performed computational experiments and analyses with oversight from G.A.C.; W.J.L. synthesized transition state mimic-containing oligonucleotides; C.H.T.A., S.S.D., A.J.F., G.A.C., and M.P.H. analyzed data and wrote the paper. All authors provided comments on the manuscript.

## Competing interests

The authors declare no competing interests.
