## [Transparent Peer Review file · Nature Communications]

Structure of human MUTYH and functional profiling of cancer-associated variants reveal an allosteric network between its [4Fe-4S] cluster cofactor and active site required for DNA repair

Corresponding Author: Professor Sheila David

Version 0:

Reviewer comments:

Reviewer #1

(Remarks to the Author)

This manuscript by the David/ Horavath/ Cisneros laboratory highlights an essential role for the iron-sulfur cluster in Human MUTYH during repair of OG:A lesions in the DNA. This work constitutes a comprehensive structure-function analysis of cancer related variants in the vicinity of the Fe-S cluster. The manuscript is well-written, the experiments conducted are rigorous with appropriate controls and the structural data are sound. My recommendation is accept with very minor revisions. I have a few minor concerns listed below:

1. The electron density shown in figure 1 is hard to see. Perhaps a change in color or the way the figure was generated could be altered to make this clearer. Furthermore, is it possible to show a difference map or a composite omit map instead of a 2Fo-Fc map?
2. Why is there such a discrepancy in the ICP-MS data in table 1 between the WT and MBP free WT. More details in the text as to how to interpret this and such type of data would be useful and make the section more clear.
3. Some of the figures/ figure panels are displayed but not discussed. For example there is no Figure 2F or 3A-C discussed in the text. Furthermore, there is no text about Figure 7!

Reviewer #2

(Remarks to the Author)

Trasvina-Arenas and colleagues report X-ray crystal structures and biochemical analysis of a human MUTYH DNA complex and characterization of potential cancer associated variants. The authors report structures of both human MUTYH in complex with DNA, and the structure of a mutant bacterial MutY. The data appear generally solid and the authors present a plausible model for how the Arg241/R149 substitution impacts MUTYH function through impacting cofactor engagement. However, much of this manuscript mirrors the authors previous work on other MUTYH mutations. Thus there are limited novel conceptual advances put forth in this manuscript. There are existing structures of mouse MUTYH and the author's previous structural and biochemical analysis of related mutations that impact the enzyme in analogous ways (Mutant N146S, Demir et al, reference 26) take away from the impact of this study. In this sense, the paper is probably better suited for a specialized journal.

Points:

1. Line 41. Word choice "famous", perhaps "best characterized" would be more appropriate?
2. Line 56 and throughout text. The authors use cysteinyl rather than cysteine?
3. Line 101-104. This text could be moved to the Methods and Materials
4. Line 119. X-ray absorption spectra could be included in the manuscript
5. Line 142. The authors state that Arg241 is hydrogen bonding with the carbonyl amide of Cys 290? This should be

corrected to main chain carbonyl of Cys 290?

6. Line 179 CD spectroscopy. I was surprised by the impact of the W103C substitution. Does the MBP not contribute to the CD signal at all? More explanation is needed here. The authors should also include SDS-PAGE gels of the purified proteins.

7. Line 213-214. This passage is confusing, at least to this referee. Please consider rewording

8. Line 257 – should be figure 2d

9. Line 268. What was done to verify the ligand geometry for Ca²⁺ to verify identity of the metal? Could the metal be inducing the conformational changes, rather than the substitution?

Reviewer #3

(Remarks to the Author)

This manuscript reports new structural and biochemical characterizations of the human MUTY base excision repair enzyme with a focus on its [4Fe-4S] cluster. The role of Fe-S clusters in DNA processing enzymes remains highly enigmatic and is of wide topical interest. This study contributes important new data for one of the paradigm proteins MUTY, which in turn provides unique mechanistic insights into the coupling of the [4Fe-4S] cluster redox to modulation of the binding of its DNA substrates. The experiments are well designed and the data are properly interpreted, nicely linking structure, enzymatic function, and repair function and leveraging information on cancer-associated variants that is unique to the human protein. Methods are described at level of detail sufficient to reproduce the experiments. The manuscript is well written- clear and easy to follow.

There is a single concern: a problem with the number of significant figures in the values of parameters cited for the measurements shown in Figures 4 and 5. The inclusion of 3, 4, and even 5 significant figures for some of the values reported is difficult to understand given the nature of the measurements and computational predictions. Large numbers of significant figures must be fully justified or the values adjusted, taking into consideration differences between fitting errors, variability of independent replicates, and intrinsic precision of the measurement/prediction.

Editorial suggestions (listed by line number)

30- 'solved' should be replaced by determined to be grammatically correct. The phase problem is solved but structures are determined.

33- replace 'show' with support a model in which

35-36- regulate its DNA repair function.

41- replace 'famous' with well-studied

100- replace solved with determined

139- ... hydrogen OF the side chain ...

173- ... exhibited ONLY background ...

200- THE R309C variant

324- ... in WT is through

327- replace 'takes place' with 'is found'

329- replace 'establishes' with predicts

330- replace 'function' with 'regulator'

334- ...MUTYH structure, in a complex ...

Version 1:

Reviewer comments:

Reviewer #1

(Remarks to the Author)

All of my previous concerns have been addressed strengthening the data presented and the overall flow of the manuscript. I have no further concerns.

Reviewer #2

(Remarks to the Author)

The authors have addressed the points.

Reviewer #3

(Remarks to the Author)

The authors have done an excellent job in responding to the comments raised in my review and those of the other reviewers.

Point-by-Point responses to reviewers

REVIEWER COMMENTS (black)

Our response (blue)

Reviewer #1 (Remarks to the Author):

This manuscript by the David/ Horvath/ Cisneros laboratory highlights an essential role for the iron-sulfur cluster in Human MUTYH during repair of OG:A lesions in the DNA. This work constitutes a comprehensive structure-function analysis of cancer related variants in the vicinity of the Fe-S cluster. The manuscript is well-written, the experiments conducted are rigorous with appropriate controls and the structural data are sound. My recommendation is accept with very minor revisions. I have a few minor concerns listed below:

1. The electron density shown in figure 1 is hard to see. Perhaps a change in color or the way the figure was generated could be altered to make this clearer. Furthermore, is it possible to show a difference map or a composite omit map instead of a 2Fo-Fc map?

We agree with the reviewer and appreciate the comment. In the corrected version of the manuscript, Figures 1 and 2 show the simulated annealing composite omit map in dark grey.

2. Why is there such a discrepancy in the ICP-MS data in table 1 between the WT and MBP free WT. More details in the text as to how to interpret this and such type of data would be useful and make the section more clear.

We appreciate the encouragement to further explain these results. We elaborated to explain potential reasons for the differences in Zn content as follows: "Of note, ICP-MS analysis MBP-free and MBP-fusion MUTYH showed above 4 nmol of Fe per nmol of MUTYH suggesting that forms of the recombinant proteins were fully loaded with the [4Fe-4S] cluster. In the case of Zn loading, the MBP free MUTYH contained a full complement of the cofactor (1.2 nmol of Zn per/nmol of MUTYH; Table 1), while only 20% of the MBP-fusion MUTYH population retained Zn. The MBP is located at the N-terminal end of the MUTYH protein near the [4Fe-4S] cluster and Zn binding sites. Thus, the close proximity of the MBP to the metal binding sites might cause a propensity to lose the Zn coordination during the purification; a situation that is circumvented by the immediate MBP removal after Nickel affinity chromatography for the MBP-free MUTYH purification. Indeed, the Zn site in MUTYH is more labile than the [4Fe-4S] cluster, as it was also lost during the crystallization process."

3. Some of the figures/ figure panels are displayed but not discussed. For example there is no Figure 2F or 3A-C discussed in the text. Furthermore, there is no text about Figure 7!

We thank the reviewer for pointing out these errors in appropriately calling out and discussing the figures. These errors have been corrected. We have rechecked all pointers to figures and tables for accuracy. We also reordered some portions of the text so that Figures are called out in the same order as presented in the text. In addition, portions of the figures were reordered for consecutive call-out in the text.

Some of the specific changes:

- Figure 2E (previously 2F) is now called out on page 5, line 145 and several other locations (lines 265, 375, 393, 395).
- Figures 3A-B are now called out in the section starting on page 6, line 170
- Figure 3C is mentioned on line 174.
- We added an additional call-out to Figure 3 on page 5, line 155 for emphasis.
- Figure 7 is now called out as Figure 6 in the text on page 12, line 358. The figure showing the MutY mechanism is now called out as Figure 7 on page 13, line 397.
- Figure 3C was previously Figure 2E
- Altered panel ordering in Figure 4

Reviewer #2 (Remarks to the Author):

Trasvina-Arenas and colleagues report X-ray crystal structures and biochemical analysis of a human MUTYH DNA complex and characterization of potential cancer associated variants. The authors report structures of both human MUTYH in complex with DNA, and the structure of a mutant bacterial MutY. The data appear generally solid and the authors present a plausible model for how the Arg241/R149 substitution impacts MUTYH function through impacting cofactor engagement. However, much of this manuscript mirrors the authors previous work on other MUTYH mutations. Thus there are limited novel conceptual advances put forth in this manuscript. There are existing structures of mouse MUTYH and the author's previous structural and biochemical analysis of related mutations that impact the enzyme in analogous ways (Mutant N146S, Demir et al, reference 26) take away from the impact of this study. In this sense, the paper is probably better suited for a specialized journal.

We appreciate the reviewer's observations and understand the *apparent* overlap with our previous work.

As noted by the reviewer, there is a structure of mouse Mutyh with an abasic site analog duplex. Of note, mouse Mutyh and human MUTYH share only 74% amino acid sequence identity which is quite low for accurate structure-activity correlations, especially for an enzyme whose cancer-associated mutations are present throughout the amino acid sequence. Therefore, the MUTYH structure in complex with DNA will provide an important basis for future analysis of MUTYH variants and further developing structure-function correlations for MUTYH.

With regards to our previous paper (Demir *et al.* NAR 2023), we argue that our new manuscript provides significant conceptual advance to our understanding. The idea that the metal cluster is connected to the active site in some way is an *idea* that we (and others) have previously suggested, however, in this new paper we show exactly how that connection is made. The new conceptual advance is the description of this metal-active site connection in a clear and tangible way. We are replacing speculative ideas previously suggested (by us and others) on MutY with a mechanistically accurate picture of how that connection works. We have added text to the Discussion section of the manuscript so that other readers more clearly see the novel conceptual advance. This new text reads, “These results lead to a significant conceptual advance in our understanding of this critical base excision DNA repair enzyme. Whereas we and others have speculated as to the nature of a functional connection between the [4Fe-4S] metal cluster and the active site^{18,28}, herein, we define how that connection is mechanistically established.” This addition to the discussion is provided in lines 350-352.

Additional Points:

1. Line 41. Word choice “famous”, perhaps “best characterized” would be more appropriate?

Replaced famous with “most studied” since the implication is that this lesion is the one most known to be produced by oxidative stress

2. Line 56 and throughout text. The authors use cysteinyl rather than cysteine?

Replaced cysteinyl with cysteine

3. Line 101-104. This text could be moved to the Methods and Materials

This section describing the x-ray crystallography has been shortened (now line 98-99) and the sentence describing PDB entry IDs is omitted as it also appears in the Methods.

4. Line 119. X-ray absorption spectra could be included in the manuscript

We added this data as Supplementary Figure 4 (called out line 144).

5. Line 142. The authors state that Arg241 is hydrogen bonding with the carbonyl amide of Cys 290? This should be corrected to main chain carbonyl of Cys 290?

The suggested changes have been made (line 140).

6. Line 179 CD spectroscopy. I was surprised by the impact of the W103C substitution. Does the MBP not contribute to the CD signal at all? More explanation is needed here.

We appreciate reviewer's keen observation and the encouragement to provide further rationale for these results. Discussion related to the loss of MBP CD signal in Trp103 mutants is now provided on page 7, lines 203-208. Destabilization of MBP by defects in the fusion protein partner is surprising and a phenomenon reported by others. We provide citations to these papers.

The authors should also include SDS-PAGE gels of the purified proteins.

We added in Supplementary Figure 6 the SDS-PAGE results with the purified MBP-MUTYH (line 516)

7. Line 213-214. This passage is confusing, at least to this referee. Please consider rewording

We have separated these lines into two sentences and rewritten to clarify. These lines are now 246-248 in the highlighted PDF.

8. Line 257 – should be figure 2d

Fixed. This is now on line 256, and in the re-organization of Figures is now Figure 3E. We also checked all figure call-outs to make sure they were correct in the manuscript. This resulted in some changes to figures like moving some portions of Fig 2 to Fig. 3.

9. Line 268. What was done to verify the ligand geometry for Ca²⁺ to verify identity of the metal? Could the metal be inducing the conformational changes, rather than the substitution?

The opportunity for metal binding and the conformational changes that ensue following amino acid replacement are probably linked and related; one cannot separate one from the other. We have checked the coordination geometry with the validation tool "highly coordinated waters..." in Coot which flagged this position as Ca²⁺. Although we are far from the optimal wavelength, the calcium is in a clear 4-sigma anomalous difference peak. These ideas have been added to the Methods.

Reviewer #3 (Remarks to the Author):

This manuscript reports new structural and biochemical characterizations of the human MUTY base excision repair enzyme with a focus on its [4Fe-4S] cluster. The role of Fe-S clusters in DNA processing enzymes remains highly enigmatic and is of wide topical interest. This study contributes important new data for one of the paradigm proteins MUTY, which in turn provides unique mechanistic insights into the coupling of the [4Fe-

4S] cluster redox to modulation of the binding of its DNA substrates. The experiments are well designed and the data are properly interpreted, nicely linking structure, enzymatic function, and repair function and leveraging information on cancer-associated variants that is unique to the human protein. Methods are described at level of detail sufficient to reproduce the experiments. The manuscript is well written- clear and easy to follow.

We thank the reviewer for these positive comments on our work!

There is a single concern: a problem with the number of significant figures in the values of parameters cited for the measurements shown in Figures 4 and 5. The inclusion of 3, 4, and even 5 significant figures for some of the values reported is difficult to understand given the nature of the measurements and computational predictions. Large numbers of significant figures must be fully justified or the values adjusted, taking into consideration differences between fitting errors, variability of independent replicates, and intrinsic precision of the measurement/prediction.

We thank the reviewer for pointing out these errors. We have now changed the figures appropriately.

Editorial suggestions (listed by line number)

30- 'solved' should be replaced by determined to be grammatically correct. The phase problem is solved but structures are determined.

Fixed

33- replace 'show' with support a model in which

Fixed

35-36- regulate its DNA repair function.

We have adjusted this final sentence of the Abstract to include this reviewer suggestion. This sentence now reads, "These results suggest that allosteric cross-talk between the DNA binding [4Fe-4S] cofactor and the base excision site of MUTYH regulate its DNA repair function".

41- replace 'famous' with well-studied

fixed

100- replace solved with determined

fixed- this sentence was removed in addressing shortening of this part by Rev. 2

139- ... hydrogen OF the side chain ...

fixed

173- ... exhibited ONLY background ...

fixed- now line 202.

200- THE R309C variant

fixed- line 232

324- ... in WT is through

fixed-line 340

327- replace 'takes place' with 'is found'

fixed-line 343

329- replace 'establishes' with predicts

fixed-line 344

330- replace 'function' with 'regulator'

fixed-line 346

334- ...MUTYH structure, in a complex ...

Re-worded sentence, see ine 350

Point-by-Point responses to reviewers

REVIEWER COMMENTS (black)

Our response (blue)

Reviewer #1 (Remarks to the Author):

All of my previous concerns have been addressed strengthening the data presented and the overall flow of the manuscript. I have no further concerns.

Reviewer #2 (Remarks to the Author):

The authors have addressed the points.

Reviewer #3 (Remarks to the Author):

The authors have done an excellent job in responding to the comments raised in my review and those of the other reviewers.

We thank the reviewers for their thoughtful reviews and positive comments on our manuscript!